

# Synoptic-scale drivers of the Mistral wind: link to Rossby wave life cycles and seasonal variability

Yonatan Givon*[1], Douglas Keller Jr.[2], Romain Pennel[2], Philippe Drobinski[2], Shira Raveh-Rubin[1]

1. Department of Earth and Planetary Sciences, Weizmann Institute of Science

2. Laboratoire de Météorologie Dynamique - IPSL, École Polytechnique, Institut Polytechnique de Paris, ENS, PSL Research University, Sorbonne Université, CNRS, Palaiseau France

*Correspondence to*: Yonatan Givon (yonatan.givon@weizmann.ac.il)

**Abstract.** The mistral is a northerly low level jet blowing through the Rhône valley in southern France, and down to the Gulf of Lions. It is co-located with the cold sector of a low level lee-cyclone in the Gulf of Genoa, behind an upper level trough north of the Alps. The mistral wind has long been associated with extreme weather events in the Mediterranean, and while extensive research focused on the low-tropospheric mistral and lee-cyclogenesis, the different upper-tropospheric large- and synoptic-scale settings involved in producing the mistral wind are not generally known. Here, the isentropic potential vorticity (PV) structures governing the occurrence of the mistral wind are classified using a self-organizing map (SOM) clustering algorithm. Based upon a 36-year (1981-2016) mistral database and daily ERA-Interim isentropic PV data, 16 distinct mistral-associated PV structures emerge. Each classified flow pattern corresponds to a different type or stage of the Rossby wave life-cycle, from broad troughs, thin PV streamers, to distinguished cut-offs. Each of these PV patterns exhibit a distinct surface impact in terms of the surface cyclone, surface turbulent heat fluxes, wind, temperature and precipitation. A clear seasonal separation between the clusters is evident and transitions between the clusters correspond to different Rossby wave-breaking processes. This analysis provides a new perspective on the variability of the mistral, and of the Genoa lee-cyclogenesis in general, linking the upper-level PV structures to their surface impact over Europe, the Mediterranean and north Africa.

## 1. Introduction:

The mistral is a northerly gap wind filament located at the Rhône valley in southern France. The Rhône valley separates the Massif Central from the Alpine ridge by a ~50 km wide canyon, channelling northerly winds into the Gulf of Lions (GOL) in the Mediterranean. The mistral winds yield the potential to deliver extreme weather impacts such as wildfires (Ruffault, et al.,





2017), heavy precipitation (Berthou, et al., 2014; 2018) and direct wind damage (Obermann et al., 2018). It poses a frequent threat to agriculture, and is one of the most renowned weather phenomena in France. The mistral is seen as the primary source of severe storms and Mediterranean cyclogenesis (Drobinski et al., 2005) and is recognized as the most dangerous wind regime in the Mediterranean (Jiang et al., 2003). The mistral outflow, composed of continental air-masses, picks up moisture at intense

evaporation rates over the GOL, before flowing towards the lee-cyclone in the Gulf of Genoa (see Fig. 1) and further de-stabilizing it. Indeed, precipitation response to the mistral can be seen at the Dolomite Mountains east to Italy, where the mistral outflow is often headed, and the warm front of the Genoa low is often active. Rainaud et al. (2017) related strong mistral events to heavy precipitation events occurring along the European-Mediterranean coast line. This relationship is manifested mainly by the re-moistening during mistral events, and the following flow of this moist air towards the European

mountain slopes scattered along the coast. Classified as a dry air outbreak (Flamant, 2003), the mistral brings relatively dry and cold continental air masses into the Mediterranean Sea interface, resulting in massive air-sea heat exchanges. Berthou et al. (2014) found a significant sea surface cooling in the GOL in response to strong mistral events, which in turn weakened the following precipitation event occurring in southern France. The winter mistral often reduces SST (Li et al., 2006; Millot, 1979), potentially destabilizing the water column, and indeed Schott et al. (1996) reported the initiation of deep convection in the

GOL in response to strong surface cooling generated by a severe mistral case that took place in February 18-23, 1992. This atmosphere-ocean coupling (Lebeaupin-Brossier and Drobinski , 2009), in which the mistral lowers the SST in the GOL, destabilizing the water column and potentially triggering oceanic deep convection, might be viewed as an altitude-crossing mechanism, in which anomalies from the tropopause (i.e., upper level potential vorticity anomalies) propagate down the troposphere via the mistral wind all the way to ground level, and further down, essentially to the bottom of the Mediterranean,

in cases of the onset of deep convection.

The mistral is dynamically connected to lee-cyclogenesis in the Gulf of Genoa (Drobinski et al., 2017; Guénard et al., 2005; Speranza et al., 1985; Tafferner, 1990; and others). The leading theories depicting the Alpine lee cyclogenesis process were recently reviewed (Buzzi et al., 2020). This process is characterized by an extra-tropical, baroclinic wave disturbance composed of an upper level trough (at times accompanied by a surface low ahead), propagating towards the Alps. While the

upper level trough propagates over the ~2.5 km mountain peaks of the Alps, at lower levels, the flow is blocked by the Alps, stalling the upper trough above the mountain range. A surface pressure dipole forms across the Alps, as low level cold air steadily accumulates on the wind-side of the Alps, and a mountain wake dominates the lee-side. Thus, a lee-cyclone is born, often in a phase-lock with its parent trough aloft. This initial stage of the lee cyclone is characterized with rapid deepening rates, attributed primarily to geostrophic adjustment processes. As the jet streak propagates over the Alps, the underlying mass

fluxes are deformed drastically, impairing geostrophic balance. A strong ageostrophic circulation across the orographic obstacle is generated, manifested by relatively strong upward/downward (~10 hPa/h, Jiang et al., 2003) motions in the wind/lee-side of the Alps, respectively. Both the upper trough and the surface low amplify rapidly, often forming a 'potential vorticity (PV) tower' (Čampa and Wernli 2012; Rossa et al., 2000), virtually to the point where the upper level wave amplitude



growth forces a wave-breaking. This initial stage is usually followed by a second baroclinic stage, in which the deepening rate
is dialed down (Buzzi et al., 2020). This cyclogenesis process induces, and is enhanced by, a downslope, northerly jet centered
at the Rhône valley, carrying the amounting cold air across the Alps and into the Mediterranean. This mid-level jet is the
manifestation of the mistral wind. The mistral is accelerated through the Rhône valley, and more so as the valley opens to form
a delta, apparently due to reduced surface drag (Drobinski et al., 2017). The mistral is then tilted eastward towards the Genoa
low and surges across the GOL. In his study, Tafferner (1990) demonstrated the decisive impact of topography on Genoa lee-
cyclogenesis using numerical models, and established that in the absence of topography, the Genoa cyclone may not form at
all, and is instead replaced by a slowly-deepening surface low further to the east. He also points out that the main role of the
jet-streak, recognized primarily as the Tramontane and Mistral winds, is in advecting high-PV air masses into the cyclogenesis
region. One can presume that in the absence of topography, northerly flow in the GOL is not expected to accelerate as much,
and mistral wind is expected to dramatically weaken. Mattocks and Bleck (1986) designed a numerical QG experiment, and
separately examined the role of topography and of the upper level PV anomaly in lee-cyclogenesis, and established that both
are necessary in order to produce the rapid deepening stage of the lee-cyclogenesis reported by observations. Tsidulko and
Alpert (2001) also examined the different contributions of topography and vorticity advection to Alpine cyclogenesis, and
emphasized the synergetic nature of the PV-mountain interaction crucial for reproducing observed deepening rates. Thus, the
mistral is understood as an integral part of the Alpine cyclogenesis process, making its connection to topography crucial. In
the work of Guénard et al. (2005), the mistral structure appears to vary with response to gravity wave breaking, and a thematic
separation between deep and shallow mistral is suggested, each responding to a different hydraulic state of the flow, ranging
between flow splitting, mountain wake and gravity wave breaking. Furthermore, hydraulic jumps have been diagnosed at the
upper edge of the boundary-layer during mistral events (Drobinski et al, 2001 a&b; Jiang et al., 2003), adding a strong boundary
layer impact to the complicated cyclogenesis picture.

Past mistral-related studies are mostly localized (Drobinski et al., 2001a & b; Guénard et al., 2005), case-focused (Buzzi et al.
2003; Jiang et al 2003; Plačko-Vršnak et al., 2005), or very generalized (Buzzi et al.; 1986; Smith 1986; Tafferner 1990).
However, little is known about the large-scale or synoptic state that induces, maintains and ends the mistral, beyond the mere
presence of an upper level trough and a surface cyclone south of the Alps. The spatial and temporal variability within the
mistral period remains poorly predicted, as is the mistral intensity and duration (Guénard et al., 2005). A systematic
climatological classification of the large scale flow patterns associated with mistral events has, to the best of our knowledge,
never been attempted. Therefore, this study is designed to address the following questions:

(i) Which upper-tropospheric features occur over the North East Atlantic and Europe during mistral days? Can they be
classified into a coherent sequence of distinct, reccurring, dynamically significant flow patterns?

(ii) How do the mistral upper-tropospheric flow types vary throughout the year?

(iii) Does the impact on the hydrological cycle vary according to the upper-tropospheric flow type?





(iv) Are there typical life cycles of the flow types? Are some flow types more persistent than others?

Addressing these questions will enhance our understanding of mistral variability and predictability. Here, we address these questions by classifying the regional upper-tropospheric PV distribution during mistral periods, and quantify their surface impact in terms of precipitation and surface turbulent heat fluxes. Relying on the PV perspective, we speculate that the upper-
level PV field bears the largest influence upon the mistral compared to other atmospheric parameters. Being a conservative quantity, PV can potentially reveal consistent yet unique flow patterns producing the mistral wind. Moreover, the fine-structures typical to isentropic PV surfaces allows one to identify a variety of different flow structures, as opposed to smooth wave-like formations presented by the geopotential field.

The climatological mistral database and the clustering approach are detailed in Section 2. The resulting flow types and the
corresponding impact are presented along with three illustrative cases in Section 3. The findings are then summarized in Section 4.

## 2.   Methods:

This study is based on climatological data for the 36-year period of 1981-2016. Objective mistral criteria are applied to identify mistral days and create a mistral database for its subsequent classification. The classification is performed by a self-organizing
map algorithm (SOM; see Section 2.2) applied to upper-tropospheric isentropic PV during mistral days, and is followed by a subsequent persistence-transition analysis of the resulting clusters.

**Figure 1: Geographical domains used in the analysis (See Methods section). The whole domain is used as SOM analysis input; the subregion where the Genoa cyclones are detected (CYC), and the subregion in which a northerly flow is required by the mistral definition (GOL) are marked in purple and red boxes, respectively. The topographic height from ERA Interim is shown in shading (m).**

### 2.1. Data and Mistral Criteria

The data used to objectively define mistral days was obtained by a combination of the ERA-Interim reanalysis (Dee et al., 2011) of the European Centre for Medium-range Weather Forecasts (ECMWF), and the regional climate model WRF-ORCHIDEE, downscaled from ERA-Interim, for the years 1981-2016. The WRF-ORCHIDEE model, at 20-km resolution, was performed by the IPSL Regional Climate Model (RegIPSL). The modelling framework is similar to Lebeaupin-Brossier et al. (2013) but uses the land surface model ORCHIDEE instead of DIFF (Drobinski et al., 2012; Guion et al., 2021).



To identify mistral days, first, a Genoa cyclone database is defined based on the presence of a cyclone in the CYC domain (Fig. 1) in ERA-Interim, using the sea-level pressure field at 1-degree horizontal resolution and 6-hourly time intervals. The

cyclone masks are identified in ERA Interim as the area within the outermost closed contour of the sea-level pressure field at 0.5-hPa intervals, adapted from Wernli and Schwierz (2006). Then, wind direction and speed criteria were applied to days when a Genoa cyclone is detected using the WRF-ORCHIDEE model data that allows higher resolution: NW to NE (i.e., +- 45º) wind direction at 900 hPa and 10-m wind speed of at least 2 m/s averaged in the GOL domain (Fig. 1). The objective identification yielded 2734 mistral days, comprising 21% year-round frequency, in agreement with Burlando et al., (2009).

Consecutive mistral days were grouped into mistral events. The identified mistral events duration and monthly frequency are shown in Fig. 2. Mistral events peak typically in Jan-Feb at 30% frequency, while the mistral is less frequent in summer (~10%). Most mistral events last a single day. Duration of more than 4-8 days occurs exclusively in the autumn and winter months. This distribution generally agrees with the climatological lifetime properties of the Genoa low (e.g., Campins et al. 2011).


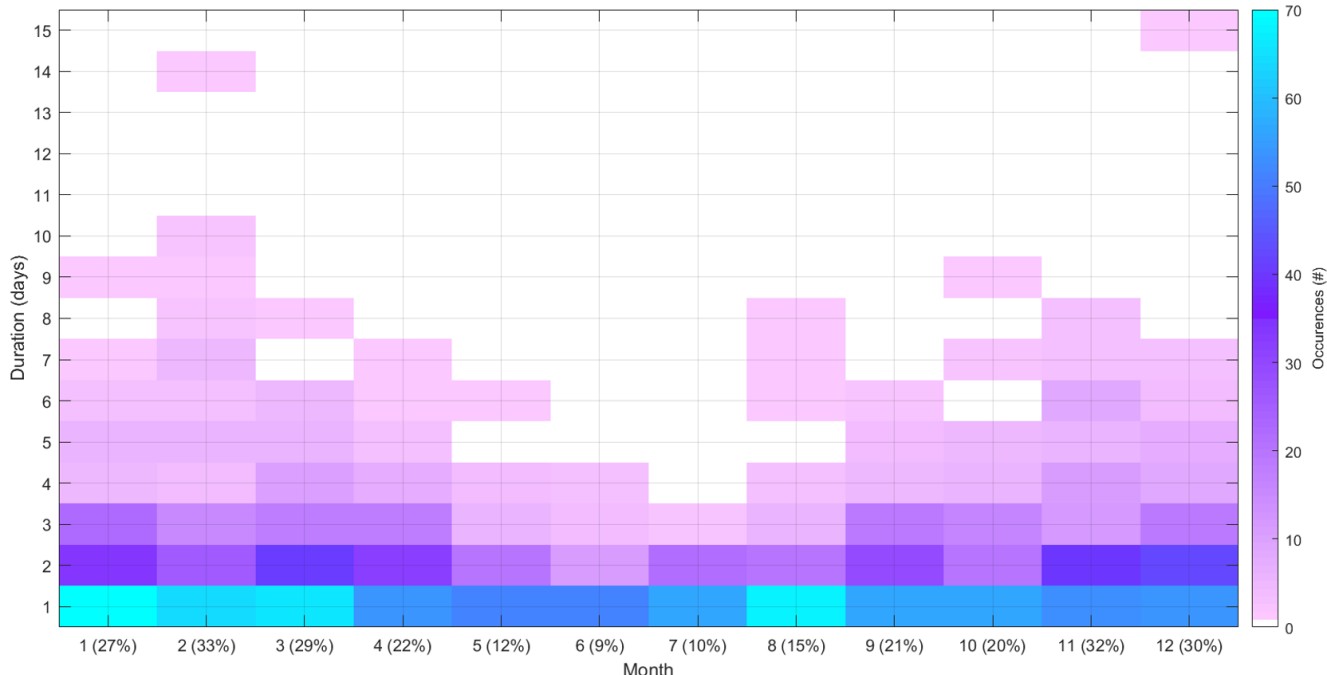

**Figure 2: Climatological monthly frequency of mistral duration (days). Shading represents the number of events within the division, for the entire time period. The percentages on the x-axis show the monthly frequency of mistral days.**



## 2.2. Isentropic PV Classification

### 2.2.1 Approach

To understand the variability of the synoptic environment during mistral, isentropic PV was classified during all identified mistral days. The classification was achieved by the self-organizing map (SOM) algorithm, reviewed by Sheridan and Lee (2011) and Liu and Weisberg (2011) in the context of synoptic meteorological classification. In contrast to more conservative clustering methods, the SOM encompasses the full continuum of the system's variability, rather than only the dominant mean states. According to Sheridan and Lee (2011), for a successful SOM clustering analysis, the variance of the analysed field can be well represented by a sequence of consistent system states, each of which corresponding to an objectively related group of samples. The choice of isentropic PV as the input field for the SOM analysis stemmed from both the natural advantages of using a conserved quantity, and its wide range of manifestations relevant for the mistral wind. Isentropic PV enables to depict fine structures such as PV streamers and cut offs attributed to Rossby wave-breaking, compared to the smoother geopotential height field. The main guidelines for a successful clustering analysis were a clear seasonal separation, a large enough number of members for each cluster, and a captured PV distribution that will emphasize the fine structures that are often averaged out of long-term PV composites. Huang et al., (2017) used a SOM algorithm combined with Hierarchical Ascendant Classification (HAC) (Jain and Dubes, 1988) to classify winter 300-K isentropic PV regimes over East Asia and related the classes to cold surges occurring in Eastern Asia. Here, the data used for the clustering analysis is the ERA-Interim daily mean vertically-averaged upper-tropospheric isentropic PV field, within the SOM domain defined by the blue box in Fig. 1. Isentropic PV is averaged between the 320-340 K isentropic levels, with intervals of 5 K. This averaging is meant to overcome the seasonal temperature variation to allow a year-round analysis (Wernli and Sprenger, 2007). The vertically averaged isentropic PV distribution is normalized by the ratio between the mean PV and the mean standard deviation, specifically:

$$PV_{norm} = PV_{(x,y,t)} * (\frac{\overline{PV}}{\overline{STD}})$$

Where the overbar denotes spatio-temporal average, thus the ratio on the right hand side is a constant equal to 0.0732, so that $PV_{norm}$ is roughly distributed between 0 and 1. Here, the SOM analysis is performed for the domain between 30-60 N and -15-30 E, a compromise between the largest domain centered at the GOL that preserved a seasonal signal, and the preserving of the identified flow patterns captured by its smaller alternatives.

The resulting clusters attribute each mistral day to one cluster. Using ERA-Interim data, we further examined the mean surface conditions for each of the resulting clusters of mistral days, to unravel the possible link between the different PV-based clusters and the surface flow, surface turbulent heat fluxes, temperature and precipitation. Transitions among clusters during mistral were then systematically examined to reveal recurring transition sequences and their seasonal variability. Finally, we demonstrate the representativeness of the clusters and transition paths to individual cases.



        **2.2.2. SOM Set-up and Validation**

The SOM Algorithm is provided by Mathworks
 (https://www.mathworks.com/help/deeplearning/gs/cluster-data-with-a-self-organizing-map.html). The setting used in the
present study yielded a shallow neural network with a single layer, using the sigmoid activation function, in hexagonal grid
(Fig. A1). The hexagonal grid setting implies that each cluster interacts with the 6 surrounding clusters in terms of similarity,
however the results are displayed on a rectangular grid to simplify the presentation. The choice of the map dimension (and
therefore number of clusters) was set as the number beyond which the classification method begins to deteriorate, or does not
add relevant information (i.e., near empty clusters or highly similar ones). Here we aimed to classify the mistral events into
clusters that pose dynamically-meaningful PV distributions, representative of their daily individual members, and with a
considerable annual mean frequency (i.e., on the order of 5%). Eventually, the number of clusters was set to 16 in a 4x4
configuration (Fig. A1), which satisfied these demands.

The SOM algorithm learning process optimizes a chosen function indicating each cluster's inner variance, and despite the wide
variety of relevant functions, well-performed SOM processes are usually quite indifferent to the chosen function (Sheridan
and Lee, 2011). This statement is usually true for other chosen parameters such as the neighborhood size and calculation time
steps, within a reasonable range of values (e.g., Cassano et al., 2006; Johnson et al., 2008). For the present study we chose the
intuitive RMSE function as the optimization parameter with the initial neighborhood distance set to 2 and number of training
steps for initial covering of the input space set to 365. Indeed, the identified patterns were only weakly affected by the choice
of these parameters. Furthermore, the SOM readily reproduced similar average patterns for several different mistral datasets.
For example, the process was repeated for a subset of mistral dates in which single-day mistral events were removed and events
separated by a single day were joined. This modified subset included 248 days less than the original, reducing the sample size
by over 10%, yet the identified patterns were nearly identical.

Statistical significance is assessed by the student's t-test between each cluster and the total averaged mistral flow. Following
Wilks et al. (2016), an additional criterion was added to the statistical test to account for the multiple testing problem.
Specifically, the Walker criteria was applied, where a threshold of $\alpha_{walker} = 1 - (1 - \alpha_0)^{N_0^{-1}}$ is set on the p-value obtained
by the t-test. $\alpha_0$ is the required significance level (e.g. 0.05 for 95% confidence) and $N_0$ the number of individual t-tests, or in
this case, the number of grid points in the domain of interest. If the p-value of an individual t-test is larger than $\alpha_{walker}$, the
null hypothesis cannot be rejected at a level of $\alpha_0$.



## 3. Results and Discussion:

### 3.1 PV Distribution Clusters

The classification of isentropic PV resulted in 16 clusters, set in a 4X4 hexagonal grid, where similar clusters are placed closer
together in the SOM space. The identified mean PV patterns defining each cluster are displayed in Fig. 3, alongside the mean
500-hPa geopotential heights. The panel order represents each cluster location in the SOM space, i.e., the least similar clusters
are placed farther apart in the SOM space, and more similar ones are placed adjacent to one another. Some clusters correspond
to exotic upper-tropospheric PV structures, such as thin streamers and cut-offs, attributed to different Rossby-wave breaking
(RWB) events. Dotted regions indicate statistical significance, i.e., the main features by which the classification process is
200 established. For instance, a high PV tongue to the west of a cut-off appears to define cluster 8, while a westward stretching PV
streamer defines cluster 9, suggesting these clusters correspond to cyclonic or anticyclonic RWB lifecycles (Thorncroft et al.,
1993), respectively. A southwesterly-oriented cut-off defines cluster 5, while a northerly thin streamer defines cluster 2, and
so on. Together, the clusters illustrate a thematic separation of the PV continuum responsible for mistral events, and one can
easily envision how the waves propagate by switching from one cluster to another.

Upstream of the high-PV anomaly, most clusters exhibit an amplified ridge in the upper troposphere over the Atlantic, a
common precursor for intense Mediterranean cyclones (Raveh-Rubin and Flaounas 2017). As expected, the primary mode of
variance is in the seasonal cycle, manifested by the meridional shift of the dynamical tropopause (subsection 3.2). Another
mode apparently picked up by the SOM is evident when comparing clusters 12 and 16 to 11 and 15. Evidently, the SOM was
able to distinguish between mountain-passed PV anomalies (11 and 15) and blocked ones (12 and 16), and the impact on the
210 trough properties is evident by the tilting of the trough axis upon the passage across the Alps from NE to N, respectively. The
standard deviation (STD) for the PV distribution within the clusters is presented in Fig. A2 in the appendix, emphasizing the
different active regions between the clusters.

Note that the composites presented on Fig. 3 each constitute ~100 days. While some variance within the clusters is inevitable,
these composites help to illustrate the SOM clustering process, especially in statistically significant regions. Detailed features
of the actual patterns, as picked up by the SOM, can be better understood by carefully examining the cluster members with
respect to their mean values. Such examples are provided in Sect. 3.6, aiding in establishing the cluster features, detailed in
Table 1.



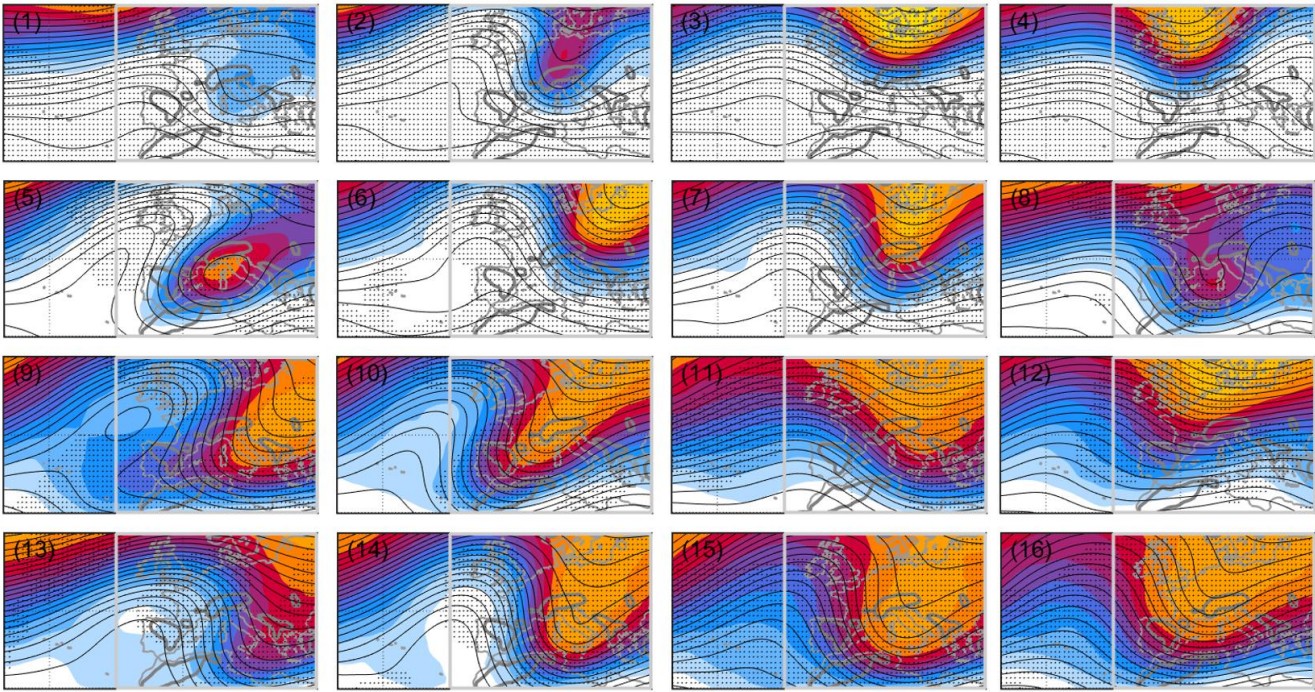

**Figure 3: Daily mean vertically-averaged isentropic (320-340K) PV (PVU, color) and 500-hPa geopotential height (unlabeled black contours, 25 m intervals) of the SOM-identified clusters. The topographic height contour of 700 m is shown in dark grey. Dotted regions indicate statistical significance >95% for the PV composites compared to the average mistral flow. The bright grey frame indicates the domain in which the SOM classification was done.**

### 3.2 Seasonal Variation

The climatological monthly occurrence frequency of each cluster is displayed in Fig. 4, demonstrating the strong seasonal affiliation of the clusters, i.e., all clusters have a clear seasonal peak in their occurrence. Clusters 1-4 occur mostly between June and October, while clusters 9-16 occur mainly between November and April. The low-PV background clearly dominates summer clusters (Fig. 3 panels 1-4) while much broader wave amplitudes constitute the winter clusters (9-16). In between, clusters 5-8 peak mainly in the transition seasons. Overall higher frequencies are obtained in the winter clusters, as expected by the larger frequency of mistral events appearing in winter.

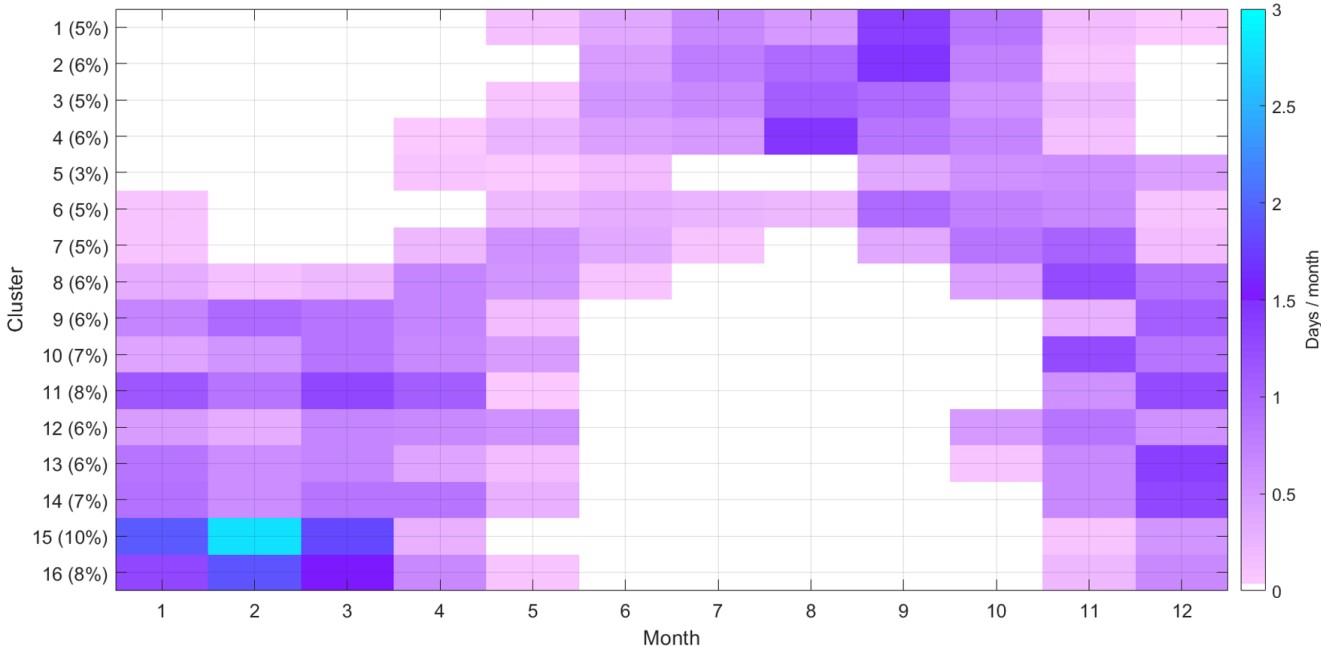

**Figure 4: Monthly frequency of the classified mistral clusters (days per month). The percentage of mistral days corresponding to each cluster is shown on the y-axis (%).**

### 3.3 Surface Circulation and Surface Impact

The surface impact of the differently classified mistral events is presented in terms composites of sea-level pressure (SLP) and precipitation (Fig. 5) and surface heat fluxes (SHF) along with 10-meter winds and 900 hPa equivalent potential temperature (Fig. 6).

The SLP patterns reveal the typical westward tilt with height, and suggest that some clusters favour a phase lock, usually corresponding to the deepest cyclones (e.g., clusters 5, 8, and 14-16). It is probable that each PV cluster is linked to a different stage of the cyclone lifecycle, which can be centered to the east or west of Italy, or even south in the Ionian Sea, with varying depths. The cyclones are closest to the lee of the Alps in clusters 4, 8, 12 and 16, associating the right column in Fig. 3 to the initial stages of cyclogenesis, while the left column (clusters 1, 5, 9 and 13) likely correspond to the termination stage of the cyclone and their easternmost location. The anticyclone extending from the Atlantic is highly variable among the clusters in its strength, thereby affecting the surface pressure gradient, and its spatial extension towards Europe. At times, the high-pressure system dominates the region (e.g., clusters 1, 6, 9, 13), such that a weak cyclone is sufficient for producing the strong mistral winds (see red arrows in Fig. 6). In other cases, the deep Mediterranean cyclone is the dominant feature (e.g., clusters 11, 12, 15, 16).

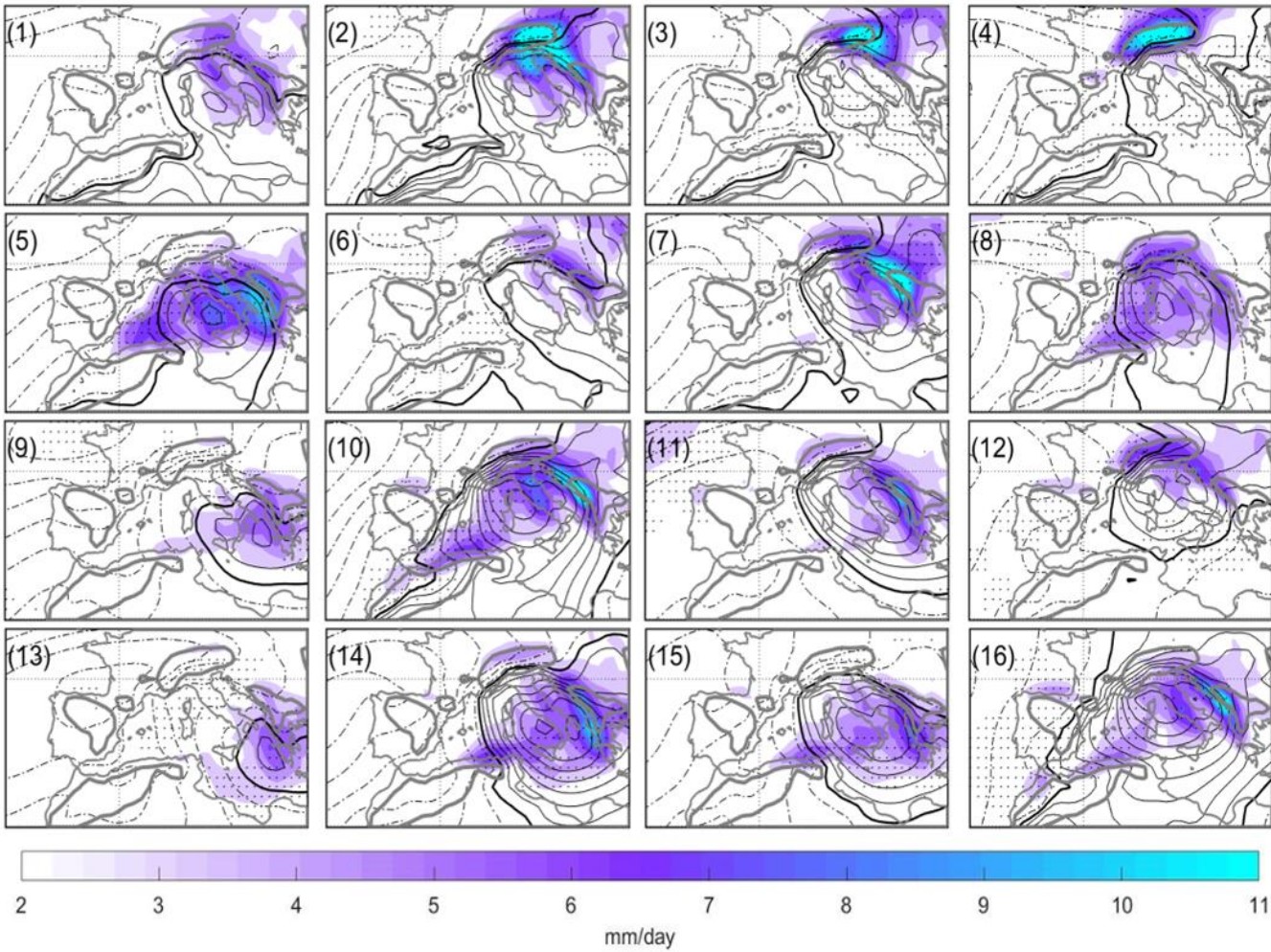

**Figure 5: Same as Fig. 3, but for a smaller domain showing SLP (hPa, black contours) at 1-hPa intervals below 1015 hPa (heavy line) and 2-hPa intervals above 1015 hPa (dotted-dashed contours). Daily accumulated precipitation (mm/day) is shaded. Dotted regions indicate statistical significance >95% for the precipitation composites compared to the average mistral flow.**

The mean distribution of precipitation is unique for every cluster, with the location of the precipitation maxima differing among clusters more than across the seasons. In summer, precipitation varies sharply between the eastern and northern Alps (i.e., clusters 1-4), while in the winter, it is differently distributed between the Dolomite and Balkan Mountains, (i.e. clusters 12, 16) and the Alps (8), with notable precipitation occurring along the African shoreline as well (10, 14, 15 and more). Generally, precipitation is distributed along the northern and eastern sides of the cyclone, and roughly correlates with its intensity, which is consistent with previous work (Flaounas et al. 2015; Raveh-Rubin and Wernli 2015, 2016).

The surface heat flux pattern associated with each cluster is relatively localized. Most clusters exhibit the familiar heat loss hotspot in the GOL, however it can extend to different lengths south into the Mediterranean and is absent from several clusters





(specifically, 8, 9, 12, and 13). The Bora winds are active together with the mistral when upper conditions allow for an easterly flow towards the Adriatic Sea in central-eastern Europe (i.e., clusters 5, 9 13 and 14), generating heat loss hotspots in the Adriatic.

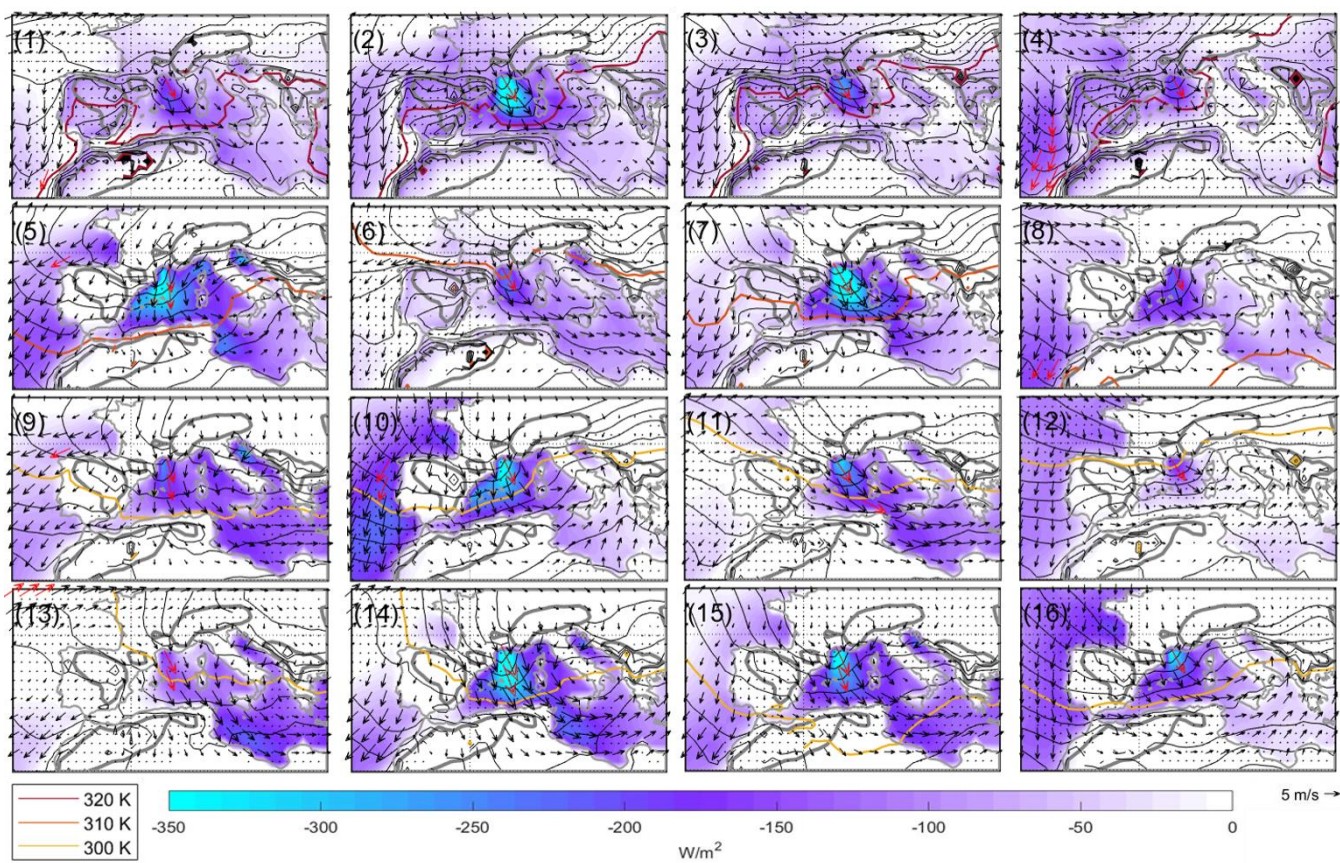

**Figure 6: Same as 5 but for surface heat fluxes (sum of sensible and latent, W m-2) in color, equivalent potential temperature at the 900-hPa level in black contours at 2-K intervals and the 300, 310 and 320-K isothermss in color (see legend). Black arrows denote 10-meter wind vectors, where red arrows mark winds above the local 75th percentile. Dotted regions indicate statistical significance >95% for the surface heat fluxes composites compared to the average mistral flow.**

The direction and horizontal extent of the mistral wind also differs between the clusters, with clusters 2, 7, 14, and 15 apparently delivering the strongest winds that also extend the furthest into the Mediterranean. The equivalent temperature field illustrates the cold and dry anomalies caused by the mistral. Differences are evident between the clusters, with some displaying a frontal deformation of the isotherms around the Gulf of Genoa. Interestingly, clusters without a marked cold/dry anomaly correspond to clusters with only weak fluxes, despite the strong surface winds (clusters 8, 9, 13).



Note that the statistical significance presented in Figs. 3, 5 and 6 is compared to the average mistral flow, thus highlighting the changes between the different mistral clusters, rather than deviations from climatology. E.g. referring to Fig. 6, the SHF maxima in the GOL are often not statistically significant as it is a standard mistral feature (clusters 6, 11 and 16). However, statistical significance arises when the signal extends further south (clusters 2, 7, and 14), west (clusters 5 and 10) or if it is exceptionally weak (clusters 1, 12, 13). The clusters main characteristics are summarized in Table 1, along with the attribution

of known mistral cases and/or Mediterranean cyclones to the PV-based clusters.

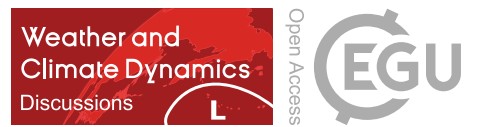

| Cluster Name | PV feature location | Trough axis | Surface cyclone location and depth | Precipitation | SHF | Known example Storm/ Study, date |
|---|---|---|---|---|---|---|
| 1. Summer cut-off | Northern Italy | N/NE | Tyrrhenian, shallow | E Alps-Dinaric, weak | Off-shore GOL, mild | Medicane Querida, 26-27/9/2006, Moscatello et al.(2008) |
| 2. Summer streamer | Northern Europe | N | Ligurian/Adriatic, deep | E Alps-Dinaric, strong | GOL, strong | Medicane Qendressa, 6/11/2014, Bouin and Lebeaupin-Brossier (2020); Cioni et al. (2018) |
| 3. Scandinavian trough | Scandinavia | N/NE | Adriatic, medium | E Alps, strong | GOL, weak | |
| 4. North Sea trough | North sea | N/NW | Adriatic, shallow | NW Alps, strong | GOL, mild | |
| 5. Genoa cut-off | Genoa | NE | Tyrrhenian, deep | Dinaric, strong | GOL, strong | Medicane Rolf, 7-8/11/2011, Dafis et al. (2018); Ricchi et al. (2017) |
| 6. Autumn/Spring trough | NE Europe | N/NE | Ionian, shallow | Dinaric, weak | GOL, mild | |
| 7. Autumn/Spring streamer | Scandinavia | N | Adriatic, deep | N Dinaric, strong | GOL, extreme | MAP IOP 15, 06/11/1999, Guenard et al. (2005) |
| 8. CWB | Genoa | N/NW | Ligurian, deep | Broad distribution, weak | GOL, weak | Dust storm (Greece), 11-12/4/2005, Kaskaoutis et al. (2008) |
| 9. Strong AWB | Eastern Europe | E | Ionian, deep | Dinaric, weak | GOL + Adriatic, weak | Medicane 15/12/2005, Dafis et al. (2020); 24/3/1998. Flamant (2003) |
| 10. Mild AWB | Central and western Europe | NE | Ligurian, deep | Dinaric, strong | GOL, strong | Medicane 16/11/2007, Dafis et al. (2020) 05/03/1982, Speranza et al. (1985) |
| 11. Relieved trough | Scandinavia-Dinaric | N | Adriatic, deep | Dinaric, strong | GOL, mild | 15/3/2013, Drobinski et al. (2017) |
| 12. Blocked trough | Scandinavia-Alps | NE | Ligurian, deep | Alps, weak | GOL, weak | 4/11/1990, Drobinski et al. (2001a) |
| 13. weak ACWB | Greece | E | Ionian, shallow | Ionian, weak | Ionian, mild | Medicane Celeno, 15/1/1996, Pytharoulis et al. (1999) |





| 14.Sharp winter streamer | Central/Eastern Europe | N | Tyrrhenian/Ionian, deep | Dinaric, strong | GOL, extreme | 13/2/2013, Drobinski et al. (2017) |
|---|---|---|---|---|---|---|
| 15. Relieved broad trough | N. Italy | N/NE | Tyrrhenian, deep | Ionian, strong | GOL, strong | Medicane Leocossia, 25/1/1982, Ernst and Matson (1983) |
| 16. Blocked broad trough | Central Europe | NE | Ligurian, deep | Dinaric, strong | GOL, mild | 17/02/1992, Schott (1996) |

**Table 1 Summary of cluster names and main features. AWB/CWB refer to anticyclonic/cyclonic Rossby wave breaking, respectively.**

### 3.4 Time Evolution of Mistral Events

Frequent cluster transitions and cluster persistence can be visualized using the transition probability matrix (TPM, Fig. 7). Column 0 shows the likelihood of a mistral event to end at any cluster with no further transition. The strong amplitude along the shifted-main diagonal suggests that every cluster has a tendency to sustain itself, at a varying likelihood. We interpret this feature as a persistence demonstration by the algorithm, as the time-scale for the evolution and migration of the PV structures driving the mistral events are often longer than a day. The fact that the PV-based SOM is able to consistently classify consequent events of slow-developing waves under the same classification reinforces the robustness of the method, given proper SOM constraints (such as the number of clusters), as only significant (by SOM interpretation) differences in the daily fields within a mistral event can force a transition. After Espinoza et al. (2012), a transition is deemed statistically significant if its frequency exceeds the 90th percentile of the corresponding transition frequency derived from a 1000 random redistributions of the original sequence. We constructed a reference random distribution by considering all mistral days (recall that the clustering is performed only for mistral days and not for all other days). While Huang et al., (2017) set their criteria by the 95th percentile, here the 90th percentile threshold was selected, considering that single-day mistral events represent roughly 50% of mistral events, and are of less interest from the dynamical perspective. Still, post-mistral days are accounted for as eligible transitions in the random distributions, represented by the 0 column in Fig. 7.

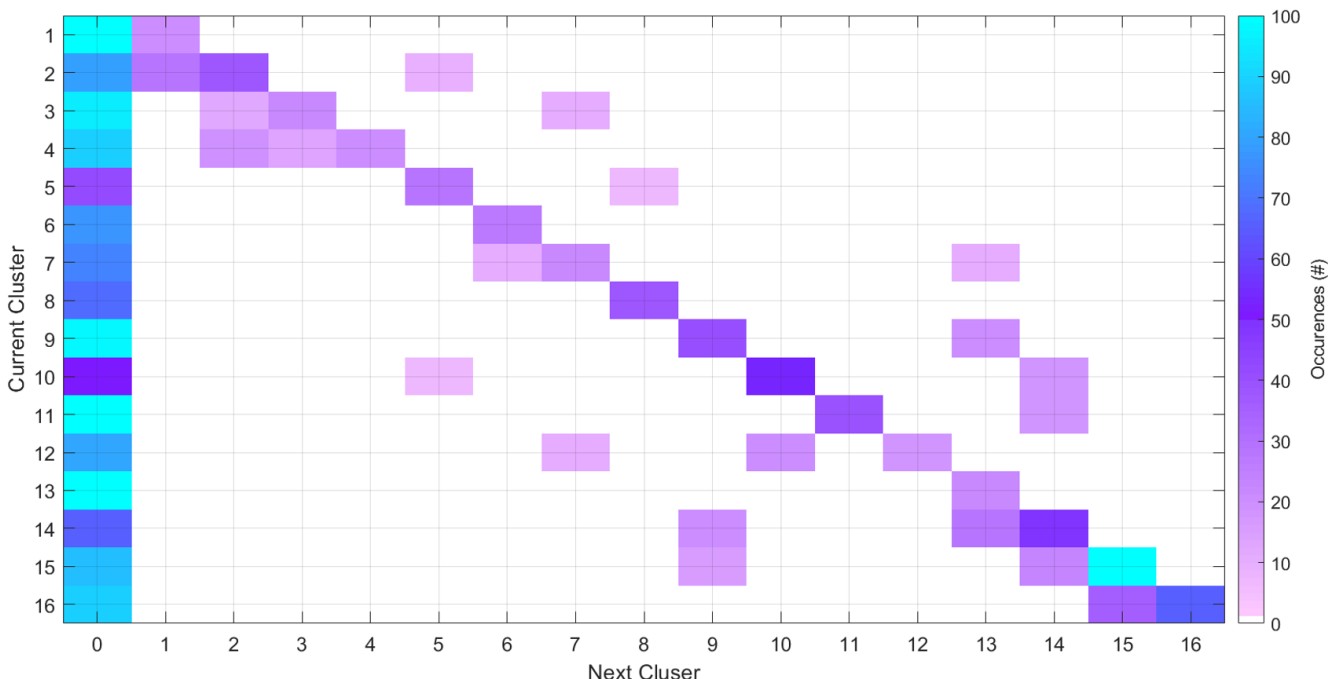

**Figure 7: Transition probability matrix, based upon daily cluster transitions from the current cluster to the cluster on the next day (colored by number of days). The 0 column represents the ending of mistral events, and the diagonal represents the probability to remain in the same cluster for the next day. Only statistically significant (90% confidence level) transitions occurring within mistral events are shown.**

Note that the transitions are distributed mostly around the main diagonal and the 0 column, the latter is primarily due to frequent single-day mistral events. Nonetheless, some recurring cluster transitions are showing a considerable amplitude, such as transitions 14→9 and 2→5, and others. These transitions are made clearer when viewed separately for each season (Fig. 8). Very different amplitudes along the main diagonal and the 0 column suggest some clusters are only self-sustaining in certain seasons and are more likely to be the end of a mistral event in the other seasons (for example, cluster 5 in autumn compared to winter-spring). It is clear from the seasonal TPMs that some transitions are absent from certain seasons, whereas others may occur at any month of the year. Studied carefully, these transitions reveal many details about the development of upper level PV anomalies over the Alps. For instance, transition 12→7 (amplifying ridge behind streamer) and 10→5 (formation of Genoa cutoff) are more abundant in the transition seasons, while transition 14→9 (strong anticyclonic Rossby wave breaking, AWB) occurs in any season except the summer, suggesting this transition and several others are more resilient to the changes of the seasons. Some of these transitions, and indeed, some individual clusters, can be directly related to AWB (14, 15 leading to 9, and 13), cyclonic Rossby wave breaking (CWB, cluster 8), a cut-off low migrating into the domain from the north (2→ 1/5), or being cut out of a north-easterly streamer (10→5). Other transitions imply an equatorward stretching of a trough (12→

10), or the eastward propagation of a trough (4→ 3, 7→ 6), and so on. These features shed light on recurring wave-evolution

processes, and allow one to easily access and investigate a large variety of rare yet dynamically similar events by selecting

certain cluster transitions that are representative of certain wave lifecycles.

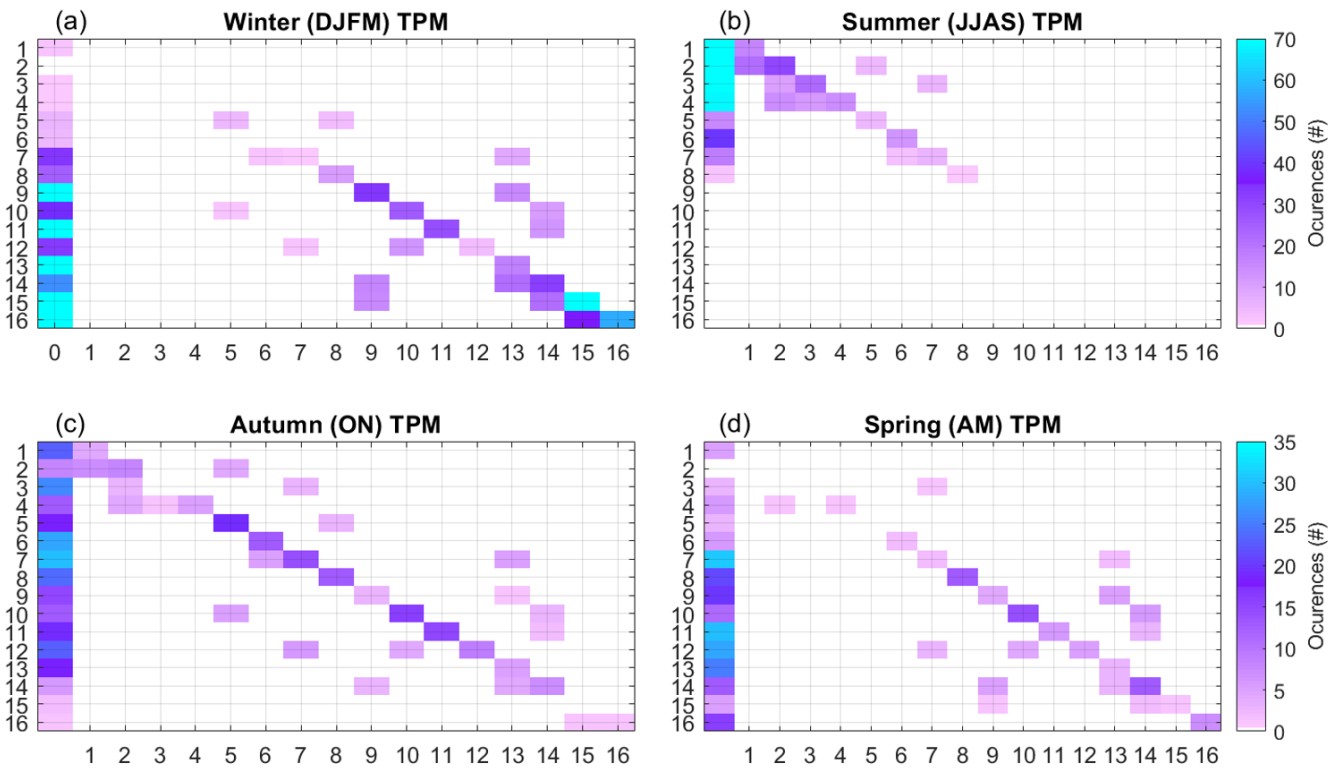

**Figure 8: The same as 7, separating by the different seasons. Note the 2-month periods in autumn (October-November) and spring**

**(April-May), compared to 4 months in winter (December-March) and summer (June-September), and the different color scales,**

**accordingly).**

Considering that many mistral events last more than a couple of days, it is insightful to examine the first event transitions,

rather than all individual transitions without a time trace (as in Figs. 7 and 8). Therefore, we first identify the mistral initiation

days, with respect to each cluster, and then display the first two transitions for every mistral event in the group. Despite

containing tens of individual transition sequences at each mistral group, several dominating first transition sequences emerge

(i.e., same sequence of days 1-3). Recurring transitions exhibit a strong seasonal dependency (Fig. 9). Qualitatively, the

transition sequences seem to "push" the system towards seasonally dependent preferable clusters. Thus, an event initiating

with an off-season cluster, say cluster 6 in winter, will be shifted towards the proper winter cluster 13, suggesting the

broadening of the northerly streamer into a mature trough. Another example can be seen in the initial clusters 4 and 8, where

the first transitions are seasonally dependent. In winter time, mistral events are dominated by the transitions between clusters




9-11 and 14-16, with preferable paths illustrated by the thick blue lines in the corresponding panels. This view also highlights the directionality of the transitions. For instance, note that the clusters with largest numbers of initiated events that last 3 or more days are the blocked clusters 12 and 16, and that the "relieved" clusters rarely jump back to a blocked cluster, i.e., the transitions 11/15→12 are scarce. Considering the cluster configuration displayed in Fig. 3, the general direction of flow within

a mistral event is from right to left and from top to bottom, diverging mostly from clusters 8 and 12 and converging towards clusters 9, 14 and 15 (see Fig. S1 in supplementary material).

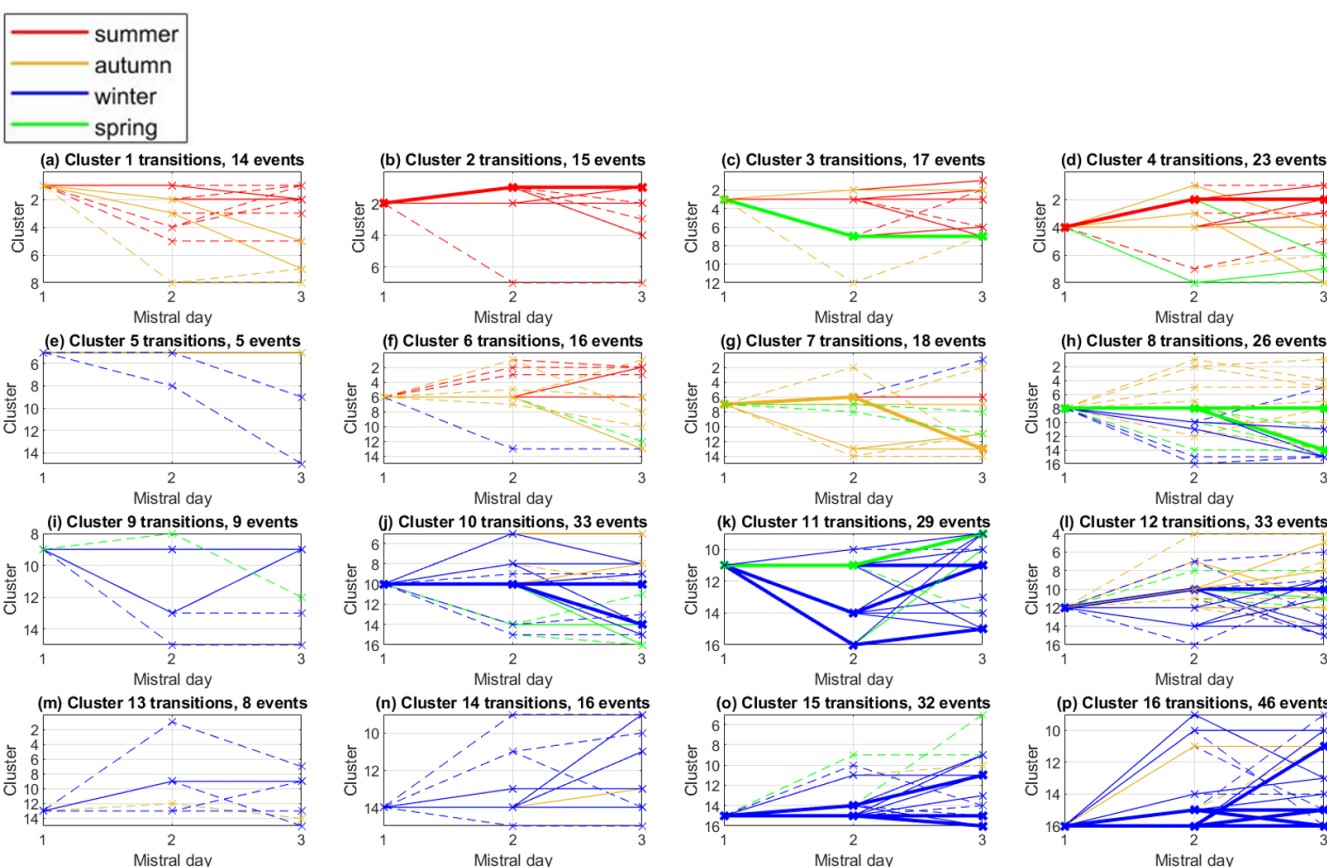

**Figure 9: Transition sequences (days 1-3) for mistral events initiating with each cluster at mistral day 1 and lasting at least 3 days,**

**at every season (using the same 2 and 4-months seasonal definitions as in Fig. 8). The x axis represents days passed from mistral initiation. Solid lines indicate transition sequences (days 1-3) that repeat 2-4 times throughout the time-period. Thick lines emphasize sequences occurring 5-10 times while dashed lines denote single-occurring sequences.**

Overall, the clustering analysis identified robust and distinct isentropic PV patterns, and the transition analysis offers additional perspective on the evolution of these upper level PV structures as they interact with the Alpine ridges during mistral events.

These TPMs, combined with the surface impact of each identified PV cluster, can potentially be utilized to improve weather



predictions of both the mistral wind and the Genoa cyclone, and their impacts. In the following we examine three individual mistral events, and the ability of the cluster mean to represent their evolution in a meaningful way.

### 3.5 Three Illustrative Mistral Events

The qualitative view of the algorithm performance is presented by a comparison between the classified clusters and the individual members for three mistral events. The large scale structure of the PV is, however, well captured by the SOM, as illustrated in the following cases.

#### 3.5.1    Anticyclonic Wave Breaking Mistral, 16-20 January 1987

This winter mistral event initiated with a broad, mountain-passed winter trough that engulfed most of continental Europe.
Strong surface heat fluxes commenced on day 1 as the trough stretched over the Alps into day 2, with the transition 15→14, accompanied by a deepening of the cyclone, and intensified precipitation and GOL heat flux, as implied by the intense mistral events generally classified in cluster 14. Another transition back to 15, from day 2 to 3, implies another broadening of the trough, with a slight weakening of the mistral and persistence in cluster 15, a common scenario (Figs. 8 and 9). The wave is then deflected to the east of the Alps and breaks anti-cyclonically, as captured by the transition 15→9. On its last day at cluster
9, the cyclone is weakened, along with reduced associated surface heat flux and precipitation, as is indeed common for cluster 9.



**Figure 10. An illustrative AWB event identified by the transition 15→9. Shown are vertically averaged (320-340 K) daily PV distributions and SLP (black contours) throughout a mistral event (middle column), alongside their corresponding cluster**





**composites (left column). Note that the view here is limited to the SOM domain (see Fig. 1). Surface impact is shown on the right**
**column in terms of the sum of sensible and latent heat fluxes (W m-2, shaded), 10-m wind vectors (the red vectors represent**
**magnitude in the upper quartile) and daily accumulated precipitation (black contours, 10 and 30 mm/day)**

### 3.5.2 Cyclonic Wave Breaking Mistral, 8-12 April 2005

This spring mistral event begins with a blocked trough and a weak cyclone in the lee of the Alps. The trough is stretched into
a thin NE streamer, captured by transition 12→10, representing a common transition. Upon this transition which marks a first
AWB, the SHF intensify dramatically, along with precipitation in the Adriatic region. The trough then further stretches and
breaks cyclonically (10→8) to form a cut-off, as the cyclone attains a deep symmetric structure (Tous and Romero, 2013).
Note that the streamer is channelled above the Rhône Valley just before breaking, illustrating the wrapping up of PV banners
generated in the mistral region (Aebischer and Schär, 1998). The classification of 11/4/2005 to cluster 8, capturing the CWB
pattern despite the PV streamer extending to the north-east rather than north-west as suggested by the composite, emphasizes
the ability of the SOM to identify distinct geometrical features, rather than only geographical ones.





**Figure 11: As in 10, but for a CWB event identified by the transition 10→8.**






### 3.5.3     Cut-off Low Mistral, 25-31 August 1995

The summer mistral event is exceptionally long for the season. It begins with the weakest upper-level forcing recorded by the present analysis, with a 2→1 transition, demonstrating the cut-off of a 3-PVU northerly streamer over the Alps. The propagation of a second wave into the domain is identified as transition 1→3, and this second wave is again stretched
southward to form a summer streamer (3→2). This transition is accompanied by an intensification of the mistral, the deepening of a primary cyclone south-east of the Baltic Sea, and a lee cyclone in the Adriatic Sea. The streamer then tilts and breaks to form a cut-off low (2→5), with weakening of the mistral intensity. A noticeable characteristic is the persistent precipitation response of the days corresponding to cluster 2, centered just between the eastern Alps and northern Dolomites, as suggested by the precipitation composites (Fig. 5, cluster 2).









**Figure 12: As in 10, but for a summer cut-off formation event identified by the transitions 2→1 and 2→5**

## 4. Summary and Concluding Remarks

This study examines systematically the large- and synoptic-scale drivers of the mistral wind during 1981-2016, by classifying the isentropic PV during mistrals. Mistral days are first identified objectively in a climatological dataset based on 20-km resolution WRF-ORCHIDEE simulation forced by ERA Interim, yielding 2734 mistral days, distributed among 1360 mistral events. Mistral occurs throughout the year, but occurs more often in winter, with more multi-day events, compared to summer. A SOM clustering analysis then classified the PV distributions during mistral days, providing insight on the large-scale driving mechanisms of the mistral wind, and further served as a tool to examine different types of surface impact signatures. Referring to the questions posed in the introduction, here we summarize the main findings (see also Table 1).

(i) During objectively-identified mistral days, the daily mean, vertically averaged (320-340 K) isentropic PV distributions are classified into 16 distinct clusters according to their geometrical shapes. The emerging features vary among amplified Rossby wave patterns ahead of an Atlantic ridge. Features include troughs, PV streamers with variable orientations indicative of cyclonic or anticyclonic Rossby-wave breaking CWB/AWB, and cut-off lows.

(ii) The clustering approach distinguishes between the seasons, including the transition seasons, suggesting that a limited range of PV features prevail in each season. In summer, clusters 1-4 suggest the dominance of either a cutoff, thin streamer or a trough over Scandinavia or the North Sea. In the extended winter season, several AWB scenarios prevail, as well as broad, deep, southward-intruding troughs. In the transition seasons, troughs, streamers cut-off lows and CWB are the dominant features. Note that PV features (e.g., streamer or cutoff-low) exhibit different mean position, shape and magnitude across different seasons (see Table 1 for a detailed summary).

(iii) Each cluster indeed reveals a unique signature in terms of surface weather impact, i.e., surface cyclone and associated circulation, winds, sensible and latent heat fluxes, and precipitation. The Bora wind regime, for example, appears to co-occur with the mistral within clusters 9 and 13, which is expected from a relatively eastern PV anomaly. The latent and sensible heat loss hotspots centered over the GOL strongly vary among the clusters, as does the wind speed and the extension of the mistral wind offshore. The highly variable mean intensity, size and location of the cyclones associated with the different clusters clearly demonstrate the impact of the upper tropospheric PV distribution on the surface pressure, with most cyclones under a phase lock with a parent trough, implying the lee-cyclogenesis effect. The deepest cyclones and strongest surface heat fluxes occur with PV streamers indicative of AWB/CWB, a Genoa cutoff or a deep winter trough. The mean precipitation response strongly depends on the cluster and season, generally being correlated to the mean cyclone intensity and peaking to the north and northeast of the cyclone. At times, precipitation occurs also downstream of the mistral outflow along the north African coast, particularly in clusters with strong surface fluxes. The latter is consistent with Rainaud et al. (2016) and Berthou et al. (2018) who demonstrate the remoistening of the dry mistral airmass and its downstream precipitation impact.



(iv) The evolution of the flow types during multi-day mistral events is examined through the analysis of transitions among clusters, while searching for recurring patterns. The dominant direction of transitions corresponds to the eastward drift of the waves (7→6, 16→15), while other transitions imply the stretching of the PV feature as it approaches the Alps (3→2/7, 12→10) or the formation of a cut-off (10→5). Long mistral events (>2 days) initiate mostly with the blocked clusters 12, 16 and 10, and tend to end with AWB/CWB events, i.e., cluster 8, 9 and 13, or with a cutoff (clusters 1, 5). The prior is mainly evident by the non-uniform distribution of the mistral initiation days between clusters (Fig. 9) and the latter by the lack of transitions from these clusters (1,5,8,9,13) to other ones. Specifically, clusters 1, 8 and 13 do not transit to any other cluster (within the 90% confidence level, see Fig. 7), with clusters 5 and 9 only transiting to 8 and 13, respectively. Cluster 14 represents the strongest mistral events in terms of heat fluxes and wind speeds, while clusters 12 and 13 correspond to the weakest. Cluster 15 is the most persistent and abundant, whereas cluster 12 is the most transient.

The SOM performance is evaluated quantitatively by standard deviations within the cluster (Fig. A2) and qualitatively by comparing individual members to their corresponding cluster composites. It is evident that even highly non-linear Rossby wave breaking processes are well represented by their corresponding cluster composite means, while the inevitable case-to-case variability is manifested mainly along the high PV gradients, or in terms of the magnitude or location of the PV anomaly of cutoffs.

One primary conclusion implied by the present classification arises when attempting to examine the present clusters under the cyclone types as defined by Tibaldi and Buzzi (1978). According to the classic description of lee-cyclogenesis, the initial phase of the lee-cyclone is attributed mostly to a-geostrophic adjustment process, induced by the mountain, generating the cyclone rapid deepening rates in a still relatively barotropic environment. In the latter phase, thermal gradients are enhanced due to the advective nature of the Rossby wave, and baroclinic instability becomes the dominant mechanism influencing the cyclone, as the low-level cold-pool air mass finally flows across the topography. Carefully examined, some clusters can be related to phase 1 or 2 of the lee cyclone, according to their representative thermal gradients, and PV maxima location relative to the Alps. The so called "blocked" modes, i.e. 12 and 16, appear to correspond to a much weaker mistral event in terms of wind speeds and GOL surface heat fluxes, when compared to their "relieved" companions, 11 and 15. This suggests that the maximum mistral related surface heat fluxes are attributed to the baroclinic phase of the cyclone, rather than the triggering phase. While further confirmation of this perspective is required, it settles with the notion of the mistral as "one of many strong winds that manifest the penetration of cold air into the Mediterranean from the north" (Scorer et al., 1952), in the sense that a strong mistral indeed enables a significant penetration of polar air-masses into the Mediterranean, leading to the massive heat loss at the GOL occurring at post-blocked stages of the mistral event, rather than in the initial stages.

This aspect of the mistral and Genoa cyclogenesis as a synoptically-controlled phenomenon has the potential to improve predictions of the mistral wind and Genoa cyclogenesis events, and deepen the understanding of synoptic-scale PV interactions with topography. Furthermore, the systematic classification offers new insight into the variability of the mistral impact on air-



sea interaction in the region, with direct implications for understanding the seasonal buildup and onset of deep convection in
the water column in autumn-winter.

**Author contribution**

YG conducted the analysis and wrote the initial manuscript draft and SRR supervised the research. SRR and PD conceptualized
the research and acquired funding, while all co-authors collaborated on the interpretation of the results and reviewing of the
manuscript.

**Acknowledgements**

This work is a contribution to the HyMeX programme (www.hymex.org) (Drobinski et al., 2014) and the Med-CORDEX
initiative (www.medcordex.eu) (Ruti et al., 2016). This article is based upon work from COST Action CA19109 "European
network for Mediterranean cyclones in weather and climate" supported by COST (European Cooperation in Science and
Technology, www.cost.eu). We thank Vered Silverman for providing the cyclone mask data. The Israel Meteorological Service
and ECMWF are acknowledged for providing access to ERA-Interim data. This work is supported by a research grant from
the Benoziyo Endowment Fund for the Advancement of Science and from the Weizmann - CNRS Collaboration Program.

**Appendix:**

A.1 The 4x4 hexagonal SOM map

The SOM neighbor distance is a measure of similarity between neighboring SOM clusters, specifically the mean squared error
(MSE) between the SOM weights corresponding to the Neuron that represents each cluster. The 4x4 hexagonal SOM map and
the neighbor distances are presented in Fig. A1. The smaller the neighbor distance the more similar the neighboring clusters.
This measure does not involve the frequency of transitions between clusters, discussed in section 3.4. The similarity map
exhibits a dark line of reduced similarity crossing the grid, defining the seasonal separation discussed in section 3.2. With that
said, there is a physical reason for similar clusters to appear consecutively, and a transition between two very different clusters,
e.g. 8→3, would seem very unlikely. Thus, the transition-season clusters are expected to provide a corridor through which
summer clusters can develop into winter clusters, or vice versa (especially cluster 7).





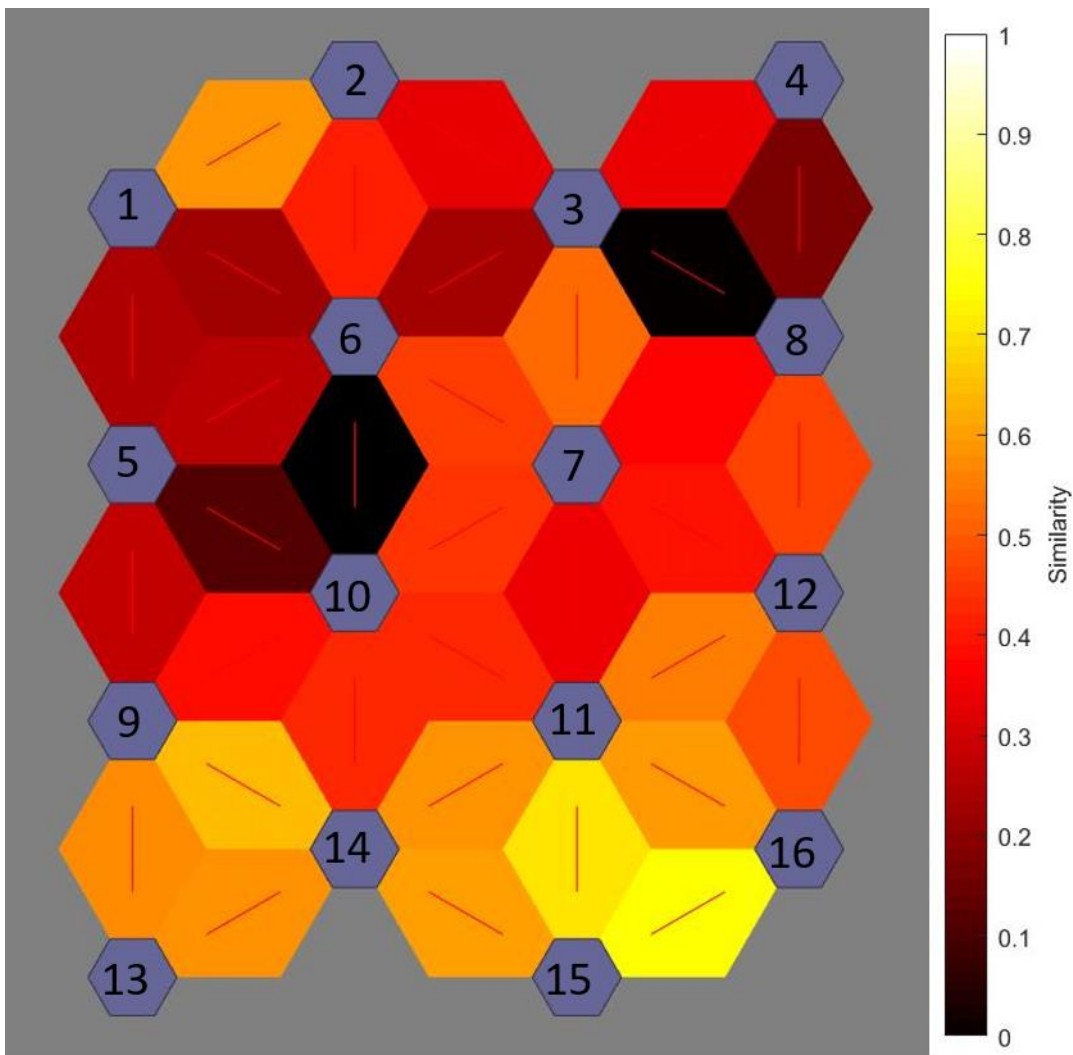

**Figure A1: The 4x4 hexagonal SOM map with the numbered clusters and neighbor distances, or similarity between clusters (dimensionless) in colored connections. Brighter colors represent short distances, or more similar clusters, and dark colors indicate**
**large distances, or dissimilarities.**

A.2 Intra-cluster variability

The variability among the members for each cluster can be quantified with the standard deviation (STD) map of the PV fields (Fig. A2). The resulting patterns demonstrate the uncertainty in the magnitude and exact location of the identified PV feature. As such, the largest STD is either along the boundary of the streamer (e.g., clusters 3, 6, 10, 14) or within a cut-off (1, 5, 8). It
is apparent that some clusters mean signal is reasonably well aligned with the individual members, while others exhibit larger inner variance. For example, most members of cluster 9 fit right in the composite, as opposed to cluster 1, where the depth of





the cut-off magnitude is likely under-estimated by the averaged field, due to variations in location. The STD also reveals further details on some clusters, such as the hidden PV streamers apparently related to clusters 6 (towards the Atlas Mountains) and 3 (towards the Pyrenees). The mean STD for most clusters is on the order of 15%, however the major correspondence

between the composite and its members is along the maximum PV gradient bands, and the quantitative inaccuracy is compensated by the qualitative description of the flow (i.e., Figs. 10-12). Compared against the 95% statistical significance (Fig. 3), focusing on cluster 5, the low STD region in the middle of the cut-off is translated into statistically significant signal, while its surrounding peak in STD is translated into non-significant regions, again demonstrating the variability regarding the extent of the identified cutoff. Similar links between the STD and statistical significance are evident in clusters 2, 7 and more.




**Figure A2: Standard deviation (STD) of isentropic PV within each cluster. The spatially-averaged STD is noted at the bottom of each panel, following the cluster number.**

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
