# Peer review of "Synoptic-scale drivers of the Mistral wind: link to Rossby wave life cycles and seasonal variability"

_Weather and Climate Dynamics, 2021_

## Author Comment (AC1)

**Reply document for reviewers of:**

**Large-scale drivers of the mistral wind: link to Rossby wave life cycles and seasonal variability**

Yonatan Givon*[1], Douglas Keller[2], Romain Pennel[2], Philippe Drobinski[2], Shira Raveh-Rubin[1]

1. Department of Earth and Planetary Sciences, Weizmann Institute of Science

2. Laboratoire de Météorologie Dynamique - IPSL, École Polytechnique, Institut Polytechnique de Paris, ENS, PSL Research University, Sorbonne Université, CNRS, Palaiseau France

*Correspondence to*: Yonatan Givon (yonatan.givon@weizmann.ac.il)

General comments

We thank the reviewers for investing the time in the review of our submission, and appreciate their instructive and useful comments. We understand that several of the methodological choices we made are up for debate, and we are pleased to respond and adjust the manuscript as suggested.

In the following we provide a point-by-point response to the individual comments, the reviewers' comments are in black, and our responses in blue.

Reviewer #1

The authors investigate the different expressions of isentropic PV that govern the occurrence and role of the mistral wind using a SOM clustering approach, which reveals new aspects of the upper-level circulation impact on low level features.

The paper is beautifully written and concise, with a thorough discussion of the findings and it should be accepted for publication in Weather and Climate Dynamics. Though there are some points that in my opinion need to be clarified, mostly regarding methodological choices. I hope the authors find the following comments useful.

We thank the reviewer for the appreciation of the work and for the constructive comments.

specific comments

- In the title, why "synoptic" and not "large-scale" drivers is chosen?

  We felt that "large scale" was too vague, however we agree that "synoptic" is often used for surface systems and thus not very accurate for the present study, hence we will change the title, as suggested.

- I don't quite understand why for the identification of the mistral days the cyclone masks are firstly applied and you did not apply the wind criterion alone. Could you please elaborate on that? Additionally, did you test the sensitivity of the results on different wind speed thresholds? (lines 118-123)

  Thank you for this question which lays the basis for our work. One of the challenges in identifying the mistral wind is its relationship with the sea breeze (Drobinski et al., 2018), as both have North-South orientations. Therefore, the

35    presence of a cyclone is traditionally considered a pre-condition for a northerly wind in the GOL to be accounted for as a mistral (see Drobinski et al., 2005; Flamant, 2003; Lebeaupin Brossier et al., 2013; Burlando, 2009 and others), as it aims to filter out standard sea-breeze northerlies. Our aim was to pinpoint the arrival of the upper level PV anomaly and accompanying low-level cold air pool into the domain, which together forms the setting for both the Alpine lee-cyclogenesis and the co-located mistral wind. Indeed, using the mistral definition that involves the

40    cyclone presence, a Rossby wave was detected impinging on the Alps throughout the resulting mistral dataset.

We have looked at sensitivities to variations in wind-speed thresholds, directional definition and the cyclone requirement. The resulting mistral frequency under the different mistral definitions are reported in Figure R1. Specifically, we changed the wind speed threshold between 2, 4, 6, 8 and 10 m/s, and the opening angle threshold between 135º, 90º and 45º (i.e., opening angle about the North, such that 90º means ±45º). It is known that cyclones

45    are frequent in the region, reaching almost 50% annual frequency by our detection (blue bars in Fig. R1). Evidently, the cyclone presence requirement reduces the number of identified days significantly (see difference between the red and yellow bars) regardless of the variations in wind criteria, and consolidates the mistral frequency in the reasonable range of 20-30%. The resulting mistral frequency is particularly sensitive to the wind direction condition. Note that due to high correlation between wind direction and wind speed in the case of the mistral, the mistral

50    frequency is almost insensitive to the wind-speed threshold with a 45º opening angle about the North (i.e., +-22.5º), suggesting that purely northerly winds are also the strongest ones.

This issue will be expanded in section 2.1 in the revised article.

Burlando, M.: The synoptic-scale surface wind climate regimes of the Mediterranean Sea according to the cluster analysis of ERA-40 wind fields. Theor. Appl. Climatol. 96, 69–83. https://doi.org/10.1007/s00704-008-0033-5,

55    2009.

Drobinski, P., Bastin, S., Guenard V., Caccia, J. L., Dabas, A. M., Delville, P., Protat, A., Reitebuch, O., Werner, C.: Summer mistral at the exit of the Rhône valley. Q. J. Roy. Meteorol. Soc., https://doi.org/10.1256/qj.04.63, 2005.

Drobinski, P., Bastin, S., Arsouze, T., Beranger, K., Flaounas, E., & Stefanon, M.: North-western Mediterranean sea-breeze circulation in a regional climate system model. Clim. Dynam., 51(3), 1077-1093.

60    https://doi.org/10.1007/s00382-017-3595-z , 2018.

Lebeaupin Brossier, C., Drobinski, P., Béranger, K., Bastin, S., & Orain, F.: Ocean memory effect on the dynamics of coastal heavy precipitation preceded by a mistral event in the northwestern Mediterranean. Q. J. Roy. Meteorol. Soc., 139(675), 1583-1597. https://doi.org/10.1002/qj.2049 , 2013.

[Figure]

65

*Figure R1: mistral frequency as a function of wind criteria. The Blue columns stand for days fulfilling the cyclone criteria alone, red columns stand for days fulfilling the wind criteria alone (both speed and angle), and yellow columns are days fulfilling the combined mistral criteria of wind criteria and cyclone presence. The black rectangle marks the wind criteria in our work.*

70

- Commonly normalization refers to the use of min value and of max-min range to bring the values between 0 and 1. I would say that the choice here is more of a rescaling. Most importantly, is this step necessary for the SOM application? If not, why this modification was applied? (lines 52-56)

  We agree the term "rescaling" is more appropriate, and will change the relevant phrase.

75  The SOM algorithm is optimal for input with values between 0-1, hence some modification of the PV field is required. Generally, as mentioned in the manuscript, the SOM performance is robust with regards to considerable set-up and input modifications. However, some variations can be observed between the different scaling methodologies, with the chosen rescaling giving improved outcome. Specifically, the differences between the clusters are much clearer when the rescaling is used, as are the captured patterns themselves. To demonstrate this

80  point, in Figures R2-5 we present the same fields presented in Figures (3, 4) of the manuscript, derived from (i) no rescaling (Fig. 2,3), and (ii) standardized anomalies (Fig. 4,5), i.e. (PV- PV_mean)/STD, as suggested by reviewer #2, as input for the SOM. The identified patterns for (i) and their seasonal signal are highly similar to Figures (3-4) from the manuscript, but less sharp than the rescaled ones. The classification of the standardized anomalies (ii) yields some fairly different clusters along with similar ones. Moreover, by construction of the anomaly definition

85  using monthly mean and STD, the seasonal variability is indeed eliminated, thus hardly any seasonal signal is detected (Fig. R5). While it is arguable whether one would like to filter out the seasonal cycle or not, doing so will undoubtedly lead to an entirely different, though not less relevant, scientific discussion, than the one currently suggested by our manuscript. For our part, we consider the present decomposition of seasonal variability as one of the merits of the present method as it captures both geometrical and seasonal aspects of the upper level flow.

90  In the revised manuscript, we will elaborate on the rescaling process and its motivation in section 2.2.1

[Figure]

*Figure R2: as in figure 3 of the article, using absolute PV values and no rescaling*

95

[Figure]

*Figure R3: as in figure 4 of the article, using absolute PV values and no rescaling*

[Figure]

*Figure R4: as in figure 3 of the article, using standardized PV anomalies*

[Figure]

105    *Figure R5: as in figure 4 of the article, using standardized PV anomalies*

- The set-up and validation of SOM method are extensively discussed, though I would like to ask if the frequently used quantization and the topographical errors were examined as well.

110    Indeed, the quantization error and topographical errors were examined following Kiviluoto (1996), and are described in Table R1, both the cluster-mean and total for all clusters. These errors will be addressed in Appendix A2 of the revised manuscript.

| Cluster | 1 | 2 | 3 | 4 | 5 | 6 | 7 | 8 | 9 | 10 | 11 | 12 | 13 | 14 | 15 | 16 | Total |
|---------|------|------|------|-----|------|------|------|------|------|------|------|------|-----|-----|------|------|-------|
| QE | 2.5 | 2.4 | 2.4 | 2.6 | 2.3 | 2.3 | 2.0 | 1.9 | 2.15 | 2.0 | 2.0 | 2.0 | 2.0 | 2.0 | 2.2 | 2.3 | **2.2** |
| TE | 0.3 | 0.25 | 0.08 | 0.5 | 0.55 | 0.15 | 0.25 | 0.63 | 0.21 | 0.57 | 0.17 | 0.22 | 0.5 | 0.1 | 0.17 | 0.38 | **0.3** |

*Table R1: cluster mean Quantization error (QE) and topographical error (TE), and the average across all clusters on the rightmost column.*

Kiviluoto, K.: Topology preservation in self-organizing maps. Proceedings of International Conference on Neural
115    Networks (ICNN'96), pp. 294-299 vol.1. 10.1109/ICNN.1996.548907, 1996.

I would suggest to add the resulted PV maps for the 16 clusters along with the STD maps in the Appendix A2 in order to facilitate the reader to follow the discussion, otherwise one should go back and forth between Figure 3 and A2. In any case, I think that the resulted PV maps is necessary to be provided per se.

120 We agree that the information regarding the resulted PV maps should appear in the Appendix to facilitate the reading. In the revised manuscript we will overlay the STD maps with the PV contours.

minor comments

- In line 23, I am not sure that "filament" is the appropriate term here

  Changed to "regime".

- In lines 42-43, please define "altitude-crossing mechanism"

125 This sentence will be rephrased to read: "an altitude-crossing mechanism, namely a pathway relating different atmospheric levels, in which anomalies from the tropopause (i.e., upper level potential vorticity anomalies) modify the tropospheric flow via the mistral wind, with impacts all the way to ground level, and further down, essentially to the bottom of the Mediterranean, in cases of the onset of deep convection."

- In Figure 3, the colorbar for the PV units is missing

130 Thank you, the colorbar was added to the figure.

Reviewer #2

This analysis provides a new perspective in explaining the observed variability of Mistral events in the
135 Mediterranean recognizing different large-scale dynamical patterns in terms of PV. This work follows an approach already tested for heavy rainfall events but not yet explored for characteristics of strong winds outbreaks. In that respect is highly original. The methodology, through the use of self-organizing map clustering, is rather innovative in this field I think.

The paper is very well written, easy to follow, and right to the point. I have few remarks and I think, after having
140 improved on the following points, it will be ready for publication.

We thank the reviewer for the constructive review and for highlighting the novelty of applying the SOM method to studying mistral variability.

Specific comments:

- In line 69, " ...(1986) designed a numerical QG experiment..", the abbreviation QG has not been introduced before

145 This is now spelled out as "Quasi-Geostrophic" in the revised manuscript.

- Between lines 188 and 121. The method to identify the cyclone Era-Interim is not very clear. Could you expand on this ?

Cyclone areas are identified in ERA Interim as all the grid points found within the outermost closed contour of sea-level pressure minima (using 0.5 hPa contour intervals). This identification method produces cyclone masks,
150 essentially labelling each grid point as 'cyclone' or 'not a cyclone'. Here, to fulfil the cyclone criteria for the mistral,

we search for at least one 'cyclone' grid point within the CYC box (Fig. 1), in at least one ERA-Interim time-step (i.e., 6 hours) during the mistral day.

In the revised manuscript we will expand this description to enhance its clarity.

And, even more. how the choice of the cyclone criteria is impacting the population of the database. I guess you should mention some sensitivity here since this is directly controlling the number of the Mistral events

The cyclone identification is well established and straightforward, introduced by Wernli and Schwierz 2006 and applied in a many studies since, including for Mediterranean cyclones in Lionello et al. 2016, Raveh-Rubin and Flaounas 2017, Flaounas et al. 2018.

In the revised manuscript we will enhance the discussion of the sensitivity to the cyclone and wind criteria, see also response to Reviewer #1.

Flaounas, E., Kelemen, F. D., Wernli, H., Gaertner, M. A., Reale, M., Sanchez-Gomez, E., ... & Conte, D. (2018). Assessment of an ensemble of ocean–atmosphere coupled and uncoupled regional climate models to reproduce the climatology of Mediterranean cyclones. Climate Dynamics, 51(3), 1023-1040.

Lionello, P., Trigo, I. F., Gil, V., Liberato, M. L., Nissen, K. M., Pinto, J. G., ... & Ulbrich, U. (2016). Objective climatology of cyclones in the Mediterranean region: a consensus view among methods with different system identification and tracking criteria. Tellus A: Dynamic Meteorology and Oceanography, 68(1), 29391.

Raveh-Rubin, S., & Flaounas, E. (2017). A dynamical link between deep Atlantic extratropical cyclones and intense Mediterranean cyclones. Atmospheric Science Letters, 18(5), 215-221.

- Figure2: Any comments on the odd presence of very long duration events (14 days and 15 days). Maybe you could start drawing when the frequency is above 2 to avoid spurious counting probably due to a very low threshold on the average wind (2m/s over the GOL domain).

Thank you for noticing these special cases. Mistral events lasting over a week are a familiar phenomenon (see meteofrance article regarding the mistral wind at: http://www.meteofrance.fr/publications/glossaire/152770-mistral). Looking into the said events in Fig. 2, two events last 15 and 14 days, starting at Dec 27 1984 and Feb 1 2012, respectively. We find that the winds and the cyclones were reasonably intense throughout the duration of these events, which were both standing baroclinic lee-wave formations (as described in Smith., 1986). Therefore, these events are real and well-captured in the sense that they are not separate events that are joint together because of a weak wind day in between.

We considered removing low frequencies from Figure (2), however since the mistral definition is rather strict, we decided to leave the figure as it is and include the anomalously long events, especially as they both seem to be indeed long events that fulfil the criteria.

As explained in the response to reviewer #1, one of the challenges with identifying the mistral is its interaction with sea-breeze. By demanding the presence of a cyclone in the GOL region, we made sure that each identified mistral date indeed corresponds to lee-cyclogenesis across the Alps, the agreed-upon setting for the mistral phenomenon. Thus, even single-day, so called "spurious" events, are still representing the desired mistral phenomenon, and may well be just as significant as longer-lasting mistral events in terms of daily wind speeds, surface heat fluxes and other parameters. Figure R6 demonstrates one example of a single-day mistral with considerable surface impact, that occurred during the HYMEX IOP 13. Therefore, we think that events lasting for a single day should be included.

Smith, R. B. (1986). Further development of a theory of lee cyclogenesis. J. Atmos. Sci., 43(15), 1582-1602.

[Figure]

190

*Figure R6: as in Figures 10-12 of the article, for a representative single-day mistral event*

- Line 150 - 155: Could you clarify the normalization (which I do not fully understand in this way) ? Wouldn't it have been better to work with standardized anomalies (Pv- Pv_mean)/STD ?

Actually, classification via PV anomalies was the first thing we tried. However, we found this form of classification too geographically confined, i.e., puts more weight on the location of the anomaly rather that it's geometrical features. Thus, while suggesting some interesting insights, the use of PV anomalies failed to reproduce all the flow structures obtained by the present classification method, and it clearly does not produce the same neat seasonal separation between the different clusters. Thus we have found rescaled absolute values of PV to be more useful for this task. Please see response to Reviewer #1 including the resulting SOM clusters and their seasonality when standardized anomalies serve as input.

We will elaborate on the rescaling process and its motivation in section 2.2.1

- In Fig3 you can improve readability by plotting geopotential with a greater contour interval (every 40m ?) . In addition the colormap for PV is missing

Thank you for these suggestions. The colormap was added and the geopotential increments were increased to 30 m, as we wish to exhibit the cut-off lows in the geopotential field as well (see panel 5) which is lost when using larger intervals.

- In Fig5 the coastlines can be confused with MSLP isolines (same color). Have you tried with thin black coastlines to see if the readability improves?

The coastlines in all relevant figures were changed to thin, black lines. We hope this makes the plots clearer.

- By the way, I would change coastline color in all maps (and maybe thin out other fields as well) to improve readability of Fig.6 and case studies.

These suggestions are accepted.

- I wonder plotting the number of occurrences inside the colored rectangles would help the interpretation of the transition matrix in fig.7. It is not easy to appreciate changes in likelihood with this color scale. And the same in fig 8

Thank you for this suggestion, the numbers were added as suggested which indeed improves their readability.